# Temporal Quantitative Proteomic and Phosphoproteomic Profiling of SH-SY5Y and IMR-32 Neuroblastoma Cells during All-*Trans*-Retinoic Acid-Induced Neuronal Differentiation

**DOI:** 10.3390/ijms25021047

**Published:** 2024-01-15

**Authors:** Thomas C. N. Leung, Scott Ninghai Lu, Cheuk Ning Chu, Joy Lee, Xingyu Liu, Sai Ming Ngai

**Affiliations:** 1State Key Laboratory of Agrobiotechnology, The Chinese University of Hong Kong, Hong Kong, China; 2School of Life Sciences, The Chinese University of Hong Kong, Hong Kong, China; scottlu@link.cuhk.edu.hk (S.N.L.); chloechu829@gmail.com (C.N.C.); joylee@link.cuhk.edu.hk (J.L.); liuxy@link.cuhk.edu.hk (X.L.); 3AoE Centre for Genomic Studies on Plant-Environment Interaction for Sustainable Agriculture and Food Security, The Chinese University of Hong Kong, Hong Kong, China

**Keywords:** SH-SY5Y, IMR-32, proteomics, phosphoproteomics, neuronal differentiation, retinoic acid

## Abstract

The human neuroblastoma cell lines SH-SY5Y and IMR-32 can be differentiated into neuron-like phenotypes through treatment with all-*trans*-retinoic acid (ATRA). After differentiation, these cell lines are extensively utilized as in vitro models to study various aspects of neuronal cell biology. However, temporal and quantitative profiling of the proteome and phosphoproteome of SH-SY5Y and IMR-32 cells throughout ATRA-induced differentiation has been limited. Here, we performed relative quantification of the proteomes and phosphoproteomes of SH-SY5Y and IMR-32 cells at multiple time points during ATRA-induced differentiation. Relative quantification of proteins and phosphopeptides with subsequent gene ontology analysis revealed that several biological processes, including cytoskeleton organization, cell division, chaperone function and protein folding, and one-carbon metabolism, were associated with ATRA-induced differentiation in both cell lines. Furthermore, kinase-substrate enrichment analysis predicted altered activities of several kinases during differentiation. Among these, CDK5 exhibited increased activity, while CDK2 displayed reduced activity. The data presented serve as a valuable resource for investigating temporal protein and phosphoprotein abundance changes in SH-SY5Y and IMR-32 cells during ATRA-induced differentiation.

## 1. Introduction

The SH-SY5Y and IMR-32 cell lines are extensively utilized in vitro models for investigating diverse aspects of neuronal cell biology and physiology [1,2,3]. As cell lines derived from human neuroblastoma tumors, SH-SY5Y and IMR-32 retain the capacity for proliferation in an undifferentiated state yet can be induced to differentiate into neuron-like phenotypes through treatment with agents such as all-*trans*-retinoic acid (ATRA). Compared to primary neurons isolated from animal models, these immortalized human cell lines overcome limitations in the availability and propagation capacity of mature primary neurons. Their human origin also allows for direct study of human molecular mechanisms without potential issues from interspecies variability. Due to these advantages, SH-SY5Y and IMR-32 cells have proven invaluable for investigating neuronal differentiation [1,4,5,6,7], neurotoxicity [8,9,10], and neurodegenerative disease [3,11,12,13,14]. Although SH-SY5Y and IMR-32 cells are extensively used as in vitro neuronal models, a systematic comparison of their dynamic proteome and phosphoproteome changes throughout differentiation remains incomplete, hindering a comprehensive understanding of the molecular events underlying their differentiation and function.

Quantitative proteomic and phosphoproteomic profiling of retinoic acid-induced differentiation of SH-SY5Y and IMR-32 neuroblastoma cells could elucidate associated molecular alterations. While a limited number of transcriptomic studies [1] and analyses of specific protein modulations [4] during IMR-32 differentiation have been conducted, in-depth proteomic profiling throughout the differentiation process remains scarce. On the other hand, multiple quantitative proteomic analyses of SH-SY5Y cells have revealed extensive protein abundance changes during differentiation, consistently showing upregulation of proteins related to neuronal development, extracellular matrix organization, and cytoskeletal reorganization. Conversely, the downregulation of proteins involved in the cell cycle and DNA processing was observed in differentiated cells compared to undifferentiated precursors [5,6,7]. However, phosphoproteomic analyses examining retinoic acid-induced differentiation in SH-SY5Y and IMR-32 cells are limited. Given the established role of protein kinases in regulating neurite outgrowth, which implied that protein phosphorylation played an important role in neuronal differentiation [15], phosphoproteomic analysis may reveal additional regulatory events during neuronal differentiation. While valuable insights have been gained, more in-depth characterization of the dynamic proteome and phosphoproteome changes occurring during retinoic acid-induced differentiation of SH-SY5Y and IMR-32 cells would further enhance their utility as neuronal models.

In this study, we characterized the temporal proteomic and phosphoproteomic changes arising in SH-SY5Y and IMR-32 human neuroblastoma cells undergoing ATRA-induced neuronal differentiation. Cells were harvested at five time points (before differentiation induction as well as 12 h, 1 day, 3 days, and 5 days post-ATRA treatment), and resulting lysates were analyzed by tandem mass tag (TMT)-based quantitative proteomics and phosphopeptide enrichment coupled with liquid chromatography–tandem mass spectrometry (LC-MS/MS). This comprehensive approach facilitated the identification and relative quantification of differentially expressed proteins and phosphosites during the course of neuronal differentiation. Our data validate previous findings and provide novel insights into the molecular events directing ATRA-induced neurogenesis in vitro.

## 2. Results

### 2.1. Quantitative Proteomic Profiling Reveals Temporal Remodeling of the Proteome during Neuronal Differentiation

To estimate the extent of neuronal differentiation induced by different concentrations of ATRA in SH-SY5Y and IMR-32 human neuroblastoma cell lines, we conducted neurite outgrowth staining experiments. Cells were treated with varying concentrations of ATRA, ranging from 1 μM to 16 μM, and the extent of neurite outgrowth was evaluated. The results demonstrated that 8 μM ATRA treatment resulted in significant neurite outgrowth in both SH-SY5Y and IMR-32 cell lines, indicating successful neuronal differentiation (Appendix A). We then examined the temporal morphological changes induced by 8 μM ATRA treatment over the course of 5 days. As expected, both cell lines displayed significant neurite outgrowth after 5 days of induction with ATRA, demonstrating the progressive neuronal differentiation over time at this concentration (Appendix A). Subsequently, quantitative proteomic analyses were conducted to characterize the temporal alterations in the global proteome elicited by ATRA treatment in SH-SY5Y and IMR-32 human neuroblastoma cell lines. Cells were treated with ATRA for 12 h, 1 day, 3 days, or 5 days, and quantitative proteomic profiles were acquired at each time point (Figure 1). In total, 3840 and 4361 proteins were identified in SH-SY5Y and IMR-32 cells, respectively (Appendix A). Principal component analysis (PCA) of the quantified proteomes showed that the ATRA-treated groups and the control group could be separated from each other and formed five distinct clusters (Figure 2A). For SH-SY5Y, principal component (PC) 1 and PC 2 explained 51.0% and 20.6% of the data variance, respectively (Figure 2A); for IMR-32, PC 1 and PC 2 explained 42.1% and 23.5% of the data variance, respectively (Figure 2A). Hierarchical clustering corroborated these results, affirming the capacity of the quantitative proteomic data to differentiate the control group from the various ATRA-treated groups (Appendix A). Collectively, the data highlights considerable time-dependent proteomic shifts induced by ATRA. Specifically, in SH-SY5Y cell lines, 529, 607, 1147, and 1312 proteins were differentially expressed after 12 h, 1 day, 3 days, and 5 days post-ATRA treatment. Similarly, IMR-32 showed 361, 936, 973, and 1135 differentially expressed proteins at the same respective time points (Figure 2B; Appendix A).

### 2.2. Functional Enrichment Analysis Highlights Biological Processes Altered during Differentiation

To elucidate the biological roles of the differentially expressed proteins (DEPs), functional enrichment analysis was conducted. Upregulated and downregulated DEPs were separately subjected to enrichment analysis against the full set of identified proteins as the background in order to identify enriched gene ontology (GO) terms within the biological process domain (Appendix A). The top 10 most significantly enriched terms are presented in Appendix A. Enrichment analysis revealed that upregulated proteins were predominantly associated with neuronal and nervous system development, as well as cytoskeleton organization. In contrast, downregulated proteins were highly enriched in processes related to cell division, chaperone function, protein folding, and translation. Notably, although not within the top 10 enriched terms, analysis also indicated an association between downregulated proteins and the one-carbon metabolic process, including the GO terms: (GO:0006730) one-carbon metabolic process, (GO:0035999) tetrahydrofolate interconversion, (GO:0046654) tetrahydrofolate biosynthetic process, (GO:0046653) tetrahydrofolate metabolic process, (GO:0006760) folic-acid-containing compound metabolic process, and (GO:0046655) folic acid metabolic process (Appendix A). We subsequently examined the specific proteins related to these biological processes in greater detail.

#### 2.2.1. Differentiation Alters Expression Cytoskeletal Proteins

The GO enrichment analysis revealed an enrichment of several GO terms related to actin and microtubule organization among the upregulated proteins (Appendix A). This prompted a detailed investigation into the changes in the abundance of actin-binding proteins and microtubule-associated proteins (MAPs) during neuronal differentiation in our proteomic study (Figure 3). Specifically, the actin-binding proteins cofilin-1, neuromodulin, and brain acid soluble protein 1 (BASP1) were upregulated following differentiation in both the SH-SY5Y and IMR-32 cell lines. Additionally, MAP1B and MAP2 were upregulated upon differentiation in both cell lines. The proteomic analysis also revealed a general upregulation of eight cytoplasmic dyneins, including dynein light chain 1 (DYNLL1), cytoplasmic dynein 1 heavy chain 1 (DYNC1H1), cytoplasmic dynein 1 intermediate chain 2 (DYNC1I2), dynein light chain Tctex-type 1 (DYNLT1), dynein light chain roadblock-type 1 (DYNLRB1), cytoplasmic dynein 1 light intermediate chain 2 (DYNC1LI2), cytoplasmic dynein 1 light intermediate chain 1 (DYNC1LI1), and dynein light chain 2 (DYNLL2). While DYNC1H1 and DYNLRB1 were significantly upregulated exclusively in the IMR-32 cell line and DYNLL2 only in the SH-SY5Y line. In contrast, KIF11, which counteracts dynein-driven microtubule transport, was downregulated upon differentiation in both cell lines.

#### 2.2.2. DNA Replication and Cell Cycle Proteins Are Downregulated upon Differentiation

The proteomic analysis revealed decreased expression of several proteins involved in cell cycle progression upon differentiation. Specifically, components of the minichromosome maintenance (MCM) 2–7 complex, which functions as the replicative helicase to unwind double-stranded DNA during DNA replication [16], were downregulated following differentiation in both cell lines, with the exception of MCM5 in the IMR-32 cells (Figure 4). Additionally, proteins involved in lagging strand DNA synthesis exhibited reduced expression after differentiation in the two cell lines examined (Figure 4). These downregulated proteins included DNA ligase 1 (LIG1), flap structure-specific endonuclease 1 (FEN1), replication factor C subunit 4 (RFC4), replication factor C subunit 5 (RFC5), and proliferating cell nuclear antigen (PCNA) (Figure 4).

#### 2.2.3. Molecular Chaperones Are Downregulated in Differentiated Cells

We examined the protein expression pattern of the four chaperone families (namely small heat shock proteins (sHSPs), HSP60, HSP70, and HSP90) [17] and the co-chaperone (Appendix A). We identified 29 chaperones or co-chaperones that exhibited significant changes in their protein expression at at least one time point during the differentiation process. Among these 29 proteins, some did not demonstrate consistent trends across the two cell lines examined. For example, differentiation resulted in an upregulation of PFDN6 and HSPA14 in IMR-32, whereas it led to their downregulation in the SH-SY5Y. While some proteins showed divergent responses to differentiation in the SH-SY5Y and IMR-32 cell lines, it is noteworthy that all the eight members of the T-complex protein ring complex/chaperonin containing T-complex protein (TRiC/CCT), a subgroup of HSP60 family, displayed a consistent pattern of downregulation in the differentiated SH-SY5Y and IMR-32 cells following a 3-day ATRA induction (Figure 5A). In addition, it was observed that the key components of mitochondrial chaperones, HSPD1, HSPA9, HSPE1, and TRAP1 [18,19], were also downregulated in the differentiated cells (Figure 5B).

#### 2.2.4. Key Enzymes in One-Carbon Metabolism and Purine Synthesis Are Reduced during Differentiation

The interconnected metabolic pathways involved in one-carbon metabolism, specifically the methionine and folate cycles, play a crucial role in cellular function by providing one-carbon units for several biological processes [1]. In our experiments, we identified seven enzymes involved in the folate and methionine cycles that exhibited significant alterations in protein expression upon differentiation. Of these, five enzymes (dihydrofolate reductase (DHFR), C-1-tetrahydrofolate synthase (MTHFD1), monofunctional C1-tetrahydrofolate synthase (MTHFD1L), bifunctional methylenetetrahydrofolate dehydrogenase/cyclohydrolase (MTHFD2), and serine hydroxymethyltransferase 2 (SHMT2)) participate in the folate cycle, while two enzymes (S-adenosylmethionine synthase isoform type-2 (MAT2A) and adenosylhomocysteinase (AHCY)) are involved in the methionine cycle (Figure 6A). In general, these enzymes displayed downregulation at later time points (days 3 and 5 post-ATRA induction) in both cell lines (Figure 6B). More specifically, the folate cycle enzymes DHFR, MTHFD1L, and MTHFD2 exhibited significant decreases in expression at days 3 and 5 post-ATRA treatment relative to control in both cell lines. MTHFD1 and SHMT2 also showed declines, although to a lesser extent. Regarding the methionine cycle, MAT2A demonstrated significant reductions in expression over time in both cell lines. AHCY exhibited variable expression at early time points (12 h and day 1 post-ATRA induction), with a subsequent downregulation noted at later stages of differentiation.

A key biological process that one-carbon metabolism supports is purine de novo synthesis. Purine can be produced either through salvage pathways, via the recycling of existing nucleosides and nucleobases, or via de novo synthesis pathways, which utilize amino acids, 10-formyltetrahydrofolate, and bicarbonate to construct the purine [2]. In this study, we identified all six enzymes implicated in the synthesis of inosine monophosphate (IMP), the primary nucleotide synthesized in de novo purine synthesis (Figure 6C). These enzymes include amidophosphoribosyltransferase (PPAT), trifunctional purine biosynthetic protein adenosine-3 (GART), phosphoribosylformylglycinamidine synthase (PFAS), bifunctional phosphoribosylaminoimidazole carboxylase/phosphoribosylaminoimidazole (PAICS), adenylosuccinate lyase (ADSL), and bifunctional purine biosynthesis protein ATIC (ATIC). For both cell lines, most of these proteins show a decreasing trend in expression upon differentiation (Figure 6D). The exception is ADSL, which significantly increases after five days of ATRA induction in SH-SY5Y cells. Notably, despite an observable decrease in PAICS expression in IMR-32 cells, the downregulation did not attain significance at any given time point (Appendix A).

### 2.3. Quantitative Phosphoproteomic Profiling Reveals Alterations in Cytoskeletal Regulation and Cell Cycle Processes Following ATRA Treatment

Quantitative phosphoproteomic analysis using sequential metal oxide affinity chromatography (SMOAC) enrichment identified 12,505 phosphopeptides from 3039 phosphoproteins across all samples (Appendix A). PCA of the quantified phosphopeptides revealed distinct clustering of the ATRA-treated and control groups, with 43.3% and 8.1% of the variance in SH-SY5Y cells explained by PC1 and PC2, respectively. Similarly, 50.4% and 7.5% of the variance was explained by PC1 and PC2 in IMR-32 cells (Figure 7). Differential expression analysis uncovered 1888, 1746, 2113, and 2291 phosphopeptides in the SH-SY5Y cell line and 2160, 1842, 2016, and 2212 phosphopeptides in the IMR-32 cell line exhibiting significant abundance changes at 12 h, 1 day, 3 days, and 5 days post-ATRA treatment, respectively (Appendix A). Similar to the proteomic analysis, upregulated and downregulated differentially expressed phosphopeptides were subjected to GO enrichment. The result showed that the upregulated differentially expressed phosphopeptides are associated with the terms related to cytoskeleton organization. while both up- and downregulated phosphopeptides were associated with cell-cycle-related terms (Appendix A). Furthermore, kinase-substrate enrichment analysis (KSEA) indicated activation of several kinases, including glycogen synthase kinase-3 beta (GSK3B), cyclin-dependent kinase 5 (CDK5), mitogen-activated protein kinase 12 (MAPK12), protein kinase cAMP-activated catalytic subunit alpha (PRKACA), mitogen-activated protein kinase 11 (MAPK11), inhibitor of nuclear factor kappa-B kinase subunit alpha (CHUK), and casein kinase 2 alpha 2 (CSNK2A2), in both cell lines following ATRA treatment, while cyclin-dependent kinase 1 (CDK1), cyclin-dependent kinase 2 (CDK2), and aurora kinase A (AURKA) were inhibited (Figure 8).

## 3. Discussion

Our quantitative comparison of the proteomes and phosphoproteomes of undifferentiated and ATRA differentiated SH-SY5Y and IMR-32 cells across four time points revealed tight clustering among biological replicates with a distinct separation between each group in both hierarchical clustering and principal component analyses (Figure 2A and Appendix A). This implies that ATRA treatment significantly impacts the proteome and phosphoproteome in both lines examined. Notably, a substantial divergence was observed between day 1 and day 3 across both SH-SY5Y and IMR-32 datasets, implicating these as critical time points for ATRA-induced proteomic remodeling (Appendix A). In addition, relative protein abundance and GO enrichment analysis point to several biological processes associated with neuronal differentiation, which will be discussed in further detail in the following subsections. Furthermore, kinase-substrate enrichment analysis (KSEA) revealed several kinases that altered their activities following ATRA treatment (Figure 8). Notably, the analysis identified changes in the activities of multiple cyclin-dependent kinases (CDKs). Specifically, CDK5 exhibited increased activity, while CDK1 and CDK2 displayed reduced activity. These findings align with previous studies in mouse and neuronal stem cells that have reported similar modulations of CDK5 and CDK2 during neuronal differentiation [20,21,22]. The observed changes in CDK activities suggest that ATRA may regulate cell cycle dynamics in part through effects on these cell cycle regulatory kinases. Further investigation is warranted to elucidate the precise mechanisms by which ATRA induces specific alterations in CDK activation states. In addition to its role in inhibiting cell cycle progression during neuronal differentiation [20], Cdk5 emerges as a crucial player in the process of neuronal differentiation itself. Notably, Cdk5 exhibits its highest activity levels in terminally differentiated cells within the adult mouse brain [21]. The expression and kinase activity of Cdk5 gradually rise throughout brain development, reaching a peak at gestational day 17 in mice [21]. This temporal pattern aligns with the primary phase of neuronal differentiation in the developing cortex, indicating potential functions for Cdk5 in neurogenic processes [21]. Specifically, Cdk5 has been implicated in neurogenesis, playing a pivotal role in promoting neurite growth and axonal formation [23]. Experimental studies have revealed the co-localization of Cdk5 and its regulatory subunit, p35, with actin filaments in axonal growth cones [23]. Furthermore, the inactivation of Cdk5 in cultured neurons impedes neurite outgrowth, while the co-expression of Cdk5 and p35 enhances neurite length in these cells [23]. These findings underscore the significance of precisely regulated Cdk5 activity in influencing neuronal differentiation through its effects on neurite and axon development.

### 3.1. Cell Cycle Dynamics and Regulation of DNA Replication

The cell cycle is a highly regulated process consisting of four main phases: G1, S, G2, and M. Progression through the cell cycle is controlled by CDKs and their regulatory cyclin subunits [24]. A critical checkpoint exists at the G1/S transition where DNA replication is initiated, also known as restriction point [24]. The transition from the G1 to S phase requires CDK2 activation by M-phase inducer phosphatase 1 (CDC25A) [24,25]. The activity and stability of CDC25A are modulated by phosphorylation at Ser18, which protects CDC25A from ubiquitination and degradation [25]. In this study, we observed that all-*trans*-retinoic acid (ATRA) treatment reduced the phosphorylation of CDC25A Ser18 in both neuroblastoma cell lines tested (Appendix A). This would result in increased degradation of CDC25A, impairing its ability to activate CDK2. Inactive CDK2/cyclin E complexes would prevent the initiation of DNA synthesis, effectively blocking S-phase entry [25]. Moreover, the tumor suppressor p53 also regulates the G1/S transition by interacting with cyclin A [26]. Phosphorylation of p53 at Ser315 induces its binding to E2F transcription factors, which displaces cyclin A and allows the progression to the S phase [26]. We found that ATRA treatment consistently decreased the phosphorylation of p53 Ser315 in both cell lines. (Appendix A). This would lead to increased binding of p53 to cyclin A, sequestering cyclin A and preventing the activation of CDK2/cyclin A complexes, which further impedes S-phase entry.

Consistent with the phosphoproteomic data, our proteomic analysis also revealed that ATRA treatment downregulated several proteins involved in cell cycle progression. In particular, all the components of the MCM2-7 complex were downregulated following ATRA exposure (Figure 4). The MCM2-7 complex is a highly conserved heterohexameric assembly of six homologous subunits, MCM2 through MCM7, that plays a pivotal role in DNA replication by unwinding the DNA at the replication fork [16,27]. Prior studies have demonstrated that decreased MCM levels result in reduced proliferation rates and increased G1 and G2 populations, implying the blocking of the cell cycle [28].

In addition, during DNA replication, the lagging strand is synthesized in a discontinuous manner as short nucleotide segments termed Okazaki fragments. These fragments are then joined into a continuous strand [29,30]. In eukaryotes, the initiation of this process necessitates the enzymatic activity of DNA polymerase α (Pol α), which synthesizes the RNA/DNA initiator. Subsequently, replication factor C (RFC) facilitates the loading of PCNA at the primer–template junctions. PCNA recruitment enables the processive elongation of the primers by DNA polymerase δ (Pol δ). As Pol δ extends the primers, it displaces the initiator RNA/DNA primer with the aid of FEN1. Upon primer removal, the resulting nicked DNA intermediates are sealed by LIG1, which catalyzes phosphodiester bond formation between adjacent Okazaki fragments [29,30]. Consistent with this model, our proteomic analysis revealed decreased expression of several proteins involved in Okazaki fragment processing following ATRA exposure. These included PCNA, LIG1, FEN1, and two RFC subunits (RFC4 and RFC5) (Figure 4).

Overall, the data indicate that ATRA regulates multiple proteins involved in the G1/S transition, coordinately blocking cell cycle progression and proliferation in neuroblastoma cells. This effect would be expected to lengthen the duration of the G1 phase. Previous studies have demonstrated that the length of the G1 phase plays a critical role in determining whether a cell will proliferate or differentiate [22,31,32]. During G1, cells integrate environmental signals that influence their division mode and eventual cell fate. For instance, as neurodevelopment proceeds, neural stem cells lengthen their G1 phase and become more responsive to differentiation cues, ultimately resulting in neurogenesis and terminal cell cycle exit [22]. Taken together, the current findings suggest that ATRA induces differentiation in neuroblastoma cells, at least in part, by lengthening the G1 phase of the cell cycle. This enables cells to integrate differentiation signals and commit to specialized cell fates rather than continued proliferation. Further investigation of cell cycle regulation by ATRA may provide additional insights into its mechanisms of action in inducing differentiation in neuroblastoma.

### 3.2. Cytoskeletal Remodeling

One of the distinctive features of ATRA-induced neuroblastoma differentiation lies in the morphogenesis of neuron projections. This process relies on dynamic reorganization of the cytoskeleton, specifically involving microtubules and actin filaments, along with their associated proteins [33,34,35]. In this study, we observed the modulation of several regulators involved in cytoskeleton dynamics during differentiation. Notably, cofilin-1 (CFL1), an actin-binding protein, could be activated by dephosphorylation on the Ser-3 residue and is primarily linked with rapid F-actin depolymerization [36]. Previous evidence indicates CFL1-mediated actin turnover generates free actin monomers and barbed ends to remodel the cytoskeleton, enabling growth cone progression and neurite extension [37,38]. Our present findings reveal that CFL1 experiences upregulation and dephosphorylation at the Ser-3 residue following ATRA exposure in both IMR-32 and SH-SY5Y datasets (Figure 3 and Appendix A). This implies an increased activity of cofilin-1, which contributes to the reorganization of the actin cytoskeleton during the differentiation process.

In addition to CFL1, we identified another pair of actin-binding proteins, neuromodulin and brain acid soluble protein 1 (BASP1), that undergo upregulation during differentiation in both IMR-32 and SH-SY5Y cell lines (Figure 3). These two proteins share a structural similarity and are known to perform analogous functions in regulating actin dynamics [39]. Particularly, neuromodulin and BASP1 are recognized for their pivotal roles in modulating the actin cytoskeleton through the PI(4,5)P2-related pathway during neuronal differentiation [40].

Similarly, microtubule-associated proteins (MAPs), which play a pivotal role in regulating microtubule remodeling and nucleation during neuronal differentiation [41,42], have also shown alternation. Specifically, microtubule-associated protein 1B (MAP1B) showed upregulation and increased phosphorylation at Ser-1265 and Ser-1389 residues in both cell lines upon ATRA treatment (Figure 3 and Appendix A). Previous studies demonstrated that phosphorylated MAP1B is required for maintaining a population of dynamically unstable microtubules, which is essential for processes like neurite outgrowth and axon guidance during neuronal differentiation [43,44,45,46,47,48].

Likewise, microtubule-associated protein 2 (MAP2) displays a general upregulation in both cell lines following ATRA treatment (Figure 3), corroborating previous research [49,50]. MAP2 functions to prevent microtubule catastrophe, which is the abrupt microtubule transition from its growth phase to its shortening phase, therefore promoting net microtubule outgrowth [51,52]. MAP2 can also directly associate with F-actin to promote bundling and stabilization, thereby providing structural guidance for stable microtubule cross-linkage [53,54]. Additionally, it can bind directly to microtubules, further stabilizing the actin–microtubule dynamics and promoting net microtubule growth [52].

Furthermore, our data revealed an altered abundance of proteins involved in microtubule transport throughout neuronal differentiation. Cytoplasmic dyneins, mediating microtubule transport during neurite development [55,56], were upregulated. In contrast, kinesin-like protein 11 (KIF11, also termed kinesin-5), acting as a mechanical brake counteracting dynein-driven microtubule transport [57], was downregulated during differentiation (Figure 3). These findings collectively suggest the increased microtubule transport rates accompanying neuronal differentiation, potentially facilitated by the upregulation of dynein and downregulation of its opposing factor, KIF11.

In short, the coordinated regulation of cytoskeletal regulatory proteins, including upregulation of actin- and microtubule-binding factors and downregulation of the microtubule transport inhibitor KIF11, highlights the key role of cytoskeleton remodeling in ATRA-induced neuroblastoma cell differentiation.

### 3.3. Alterations in Molecular Chaperones

Proteins, acting as the primary agents of cellular activity, are required to fold into proper conformations to carry out their biological functions effectively. However, most of the proteins are marginally stable and prone to misfolding [58]. Misfolded proteins not only lose their functionality but also tend to aggregate, posing potential toxicity to cells [17]. To maintain the proper protein conformations, eukaryotic cells have developed various classes of chaperones. These chaperones guide nascent polypeptides toward their native conformation and assist in refolding or degrading non-native (misfolded) polypeptides [17]. Additionally, chaperone systems also play a crucial role in the process of cellular differentiation [59,60]. A previous study demonstrated that differentiation downregulates the ATP-dependent chaperonin TRiC/CCT in mouse neural stem and progenitor cells [60]. In alignment with this finding, this study observed the downregulation of all eight TRiC/CCT subunits upon differentiation in both cell lines (Figure 5A), reinforcing the previously established association between differentiation and the downregulation of this chaperonin class. In addition, our experiment revealed that differentiation also downregulates HSPD1 and HSPA9 (Figure 5B), two pivotal mitochondrial chaperones [18,19]. This downregulation is also observed in their respective co-chaperones, HSPE1 and TRAP1 (Figure 5B). A plausible explanation for the downregulation of these ATP-dependent chaperones is that differentiation necessitates a substantial expenditure of ATP. Differentiating cells may thus prioritize ATP allocation for essential differentiation processes, such as actin polymerization, rather than chaperone functions.

### 3.4. Metabolic Reprogramming of Folate, Methionine, and Purine Pathways

Furthermore, we have discovered that the interaction among folate, methionine, and purine metabolism is linked to ATRA-induced neuronal differentiation. Previous studies have highlighted that deficiencies in methionine and folate can impair neural stem cell proliferation and differentiation due to reduced cellular methylation potential and limited methionine availability for protein synthesis [61]. Although our investigation did not directly measure methionine and folate levels, the downregulation of key enzymes associated with these metabolic pathways suggests a potential decrease in metabolites derived from folate and methionine (Figure 6A,B). This downregulation could be either the cause or consequence of enzyme downregulation. Interestingly, despite reduced proliferation, successful differentiation was observed in ATRA-treated SH-SY5Y and IMR-32 neuroblastoma cells, indicating the involvement of additional factors in neuroblastoma differentiation.

Moreover, 10-formyl-THF, an intermediate in the folate cycle, serves as a one-carbon donor in purine de novo synthesis [62,63]. Given the observed downregulation of the folate cycle, we turned our attention to investigating the purine de novo synthesis pathway. In this pathway, inosine monophosphate (IMP) is the first nucleotide formed in purine de novo synthesis, which can subsequently be converted to either adenosine monophosphate (AMP) or guanosine monophosphate (GMP). The synthesis of IMP from phosphoribosyl pyrophosphate (PRPP) is catalyzed by six enzymes over ten steps [63]. This study found a general downregulation of these enzymes, with four out of the six enzymes significantly downregulated in both cell lines post-ATRA induction (Figure 6C,D). The remaining enzymes, ADSL and PAICS, showed a trend toward downregulation in IMR-32 and SH-SY5Y, respectively. Additionally, enzymes ADSS2 and GMPS, responsible for converting IMP to AMP and GMP, respectively, also exhibited downregulated expression following ATRA treatment (Figure 6D). Taken together, these findings indicate that purine de novo synthesis is downregulated in both SH-SY5Y and IMR-32 neuroblastoma cell lines after ATRA induction. A possible explanation for this downregulation is that differentiation reduces proliferation and consequently diminishes the demand for purine nucleotides.

To conclude, this study has successfully highlighted the key proteins involved in the aforementioned biological processes. However, it is important to note that our data encompasses only a fraction of the potential insights that can be derived from these experiments. While we focused primarily on phosphorylation, it is recognized that other post-translational modifications, such as *S*-palmitoylation, may also play significant roles in neuronal differentiation [64]. Therefore, future investigations should consider exploring additional post-translational modifications to obtain a more comprehensive understanding of the molecular mechanisms underlying this complex process. Nonetheless, this study compiled a comprehensive resource detailing the temporal abundance changes in thousands of proteins and phosphopeptides. These quantitative proteomic and phosphoproteomic profiles of differentiating SH-SY5Y and IMR-32 cells may serve as a useful reference for future studies utilizing these widely employed in vitro models of human neurons.

## 4. Materials and Methods

### 4.1. Cell Cultures and Treatment

IMR-32 (ATCC CCL-127) and SH-SY5Y (ATCC CRL-2266) cell lines were obtained from American Type Culture Collection (ATCC). IMR-32 was maintained in ATCC-formulated Eagle’s Minimum Essential Medium (EMEM, ATCC), and SH-SY5Y was maintained in Dulbecco’s Modified Eagle’s Medium (DMEM, Gibco, Waltham, MA, USA). Media were supplemented with 10% fetal bovine serum (Hyclone, Logan, UT, USA). All the cell lines were maintained at 37 °C in a 5% humidified CO_2_ atmosphere in a T-75 flask at a density of 300,000 cells/flask. Cells were passaged when confluency reached approximately 75% using trypsin–EDTA (0.25%) phenol red solution (Gibco, Waltham, MA, USA) for dissociation. For differentiation induction, cells were seeded in a T-25 culture flask at a density of 100,000 cells/flask and treated with ATRA (Sigma Aldrich, Saint Louis, MO, USA) dissolved in dimethyl sulfoxide (DMSO; Sigma Aldrich, Saint Louis, MO, USA) at specified concentrations and durations. The ATRA stock solution was prepared at a concentration of 0.5 μg/μL. The differentiation inducer was replenished every 2 days by changing the media. Differentiation extent was evaluated using the Neurite Outgrowth Staining Kit (Invitrogen, Carlsbad, CA, USA) according to the manufacturer’s guidelines.

### 4.2. Sample Preparation

After differentiation induction, cells were harvested at specified time points. Samples were collected in four independent experiments. Media were aspirated, and cells were washed twice with PBS. Cells were lysed with urea lysis buffer (6 M urea, 2 M thiourea, supplemented with Protease and Phosphatase Inhibitor Mini Tablets (Pierce, Thermo Fisher Scientific, Waltham, MA, USA)). Urea concentration of the samples was diluted to 1 M with 25 mM triethylammonium bicarbonate (TEAB; Sigma Aldrich, Saint Louis, MO, USA) at pH 8.5. Proteins were reduced with 5 mM tris(2-carboxyethyl) phosphine (Sigma Aldrich, Saint Louis, MO, USA) for 1 h at 60 °C, followed by alkylation with 10 mM methyl methanethiosulfonate (Pierce, Thermo Fisher Scientific, Waltham, MA, USA) for 10 min at room temperature in the dark. Alkylated proteins were digested overnight at 37 °C with sequencing grade trypsin (Promega, Madison, WI, USA) at a 1:10 trypsin:protein ratio. Tryptic digested peptides were subsequently purified using C18 spin columns (Thermo Fisher Scientific), then resuspended in 50 mM TEAB, pH 8.5. For labeling, samples were labeled with TMT10plex label reagent (Thermo Fisher Scientific, Waltham, MA, USA) in the following order: IMR-32—control, 126; 12 h, 127C; 1 day, 128C; 3 days, 129C; 5 days, 130C. SH-SY5Y—control, 127N; 12 h, 128N; 1 day, 129N; 3 days, 130N; 5 days, 131. TMT reagent was dissolved in acetonitrile (ACN) and incubated for 1 h at room temperature with samples. Equal amounts of each sample were mixed and fractionated into 8 fractions by High pH Reversed-Phase Fractionation Kit (Thermo Fisher Scientific, Waltham, MA, USA).

### 4.3. Phosphopeptide Enrichment

After TMT labeling, equivalent amounts of each sample were mixed and vacuum-dried, resulting in 1 mg of peptides. Phosphopeptides were enriched by sequential enrichment using the metal oxide affinity chromatography (SMOAC) method. First, phosphopeptides were enriched using the High-Select TiO_2_ Phosphopeptide Enrichment Kit (Thermo Fisher Scientific, Waltham, MA, USA) according to the manufacturer’s protocol. Flow-through and wash fractions were retained, pooled, and subjected to ferric nitrilotriacetate (Fe-NTA)-based phosphopeptides enrichment using a High-Select Fe-NTA phosphopeptide enrichment kit (Thermo Fisher Scientific, Waltham, MA, USA). Eluents were pooled and fractionated with High pH Reversed-Phase Fractionation Kit (Thermo Fisher Scientific, Waltham, MA, USA).

### 4.4. LC-MS/MS Analysis

Prior to LC-MS/MS analysis, the amount of peptides was estimated using a Quantitative colorimetric peptide assay (Pierce, Thermo Fisher Scientific, Waltham, MA, USA). A total of 1 µg of labeled peptide or phosphopeptide fractions were subjected to nano-LC separation utilizing Dionex UltiMate 3000 RSLC nano system using a 25 cm long, 75 µm i.d. C18 column. The columns were heated to 50 °C for all experiments. Mobile phases were 0.1% formic acid in water (buffer A) and ACN (buffer B). Peptide fractions were loaded onto the trapping column for 5 min at 100% A at 5 µL/min, then eluted through the analytical column at 300 nL/min with the gradient listed in Appendix A. Eluate was analyzed by an Orbitrap Fusion Lumos Tribrid mass spectrometer (Thermo Fisher Scientific, Waltham, MA, USA) using synchronous precursor selection MS3 (SPS MS3) or pseudoMS3 (multistage activation, MSA) for total proteome and phosphoproteome analysis respectively [65]. Details are provided in the Appendix A.

### 4.5. Proteomic Data Analysis

The mass spectra were analyzed using the Proteome Discoverer (version 2.4, Thermo Fisher Scientific, Waltham, MA, USA) with SEQUEST as a search engine. The data were searched against the UniProt Homo sapiens database (Proteome ID UP000005640). The search parameters were as follows: Precursor ion mass tolerance, 10 ppm; fragment ion mass tolerance, 0.02 Da; tryptic cleavage specificity, 2 maximum missed cleavages. For the modification setting, oxidation of methionine (+15.995 Da) was set as the dynamic modification and methylthiol of cysteine (+45.988 Da), TMT6plex of peptide N-terminal, protein N-terminal and lysine (+229.163 Da) were set as the static modification. In addition, the phosphorylation of serine, threonine, and tyrosine (+79.966) was set as the dynamic modification for the phosphoproteomic analysis. The protein-level false discovery rate was estimated by Percolator with an experimental q-value (exp. q-value) threshold set as 0.05. 

### 4.6. Statistical Analysis and Data Visualization

Principal component analysis (PCA) was performed in R and visualized with the ggbiplot package. Hierarchical clustering utilized Proteome Discoverer (version 2.4). Protein quantification relied on the built-in “Reporter ions quantifier” node of Proteome Discoverer. Proteins were considered differentially expressed if the difference was statistically significant (Benjamini-corrected *p*-values ≤ 0.05). For the total proteomic results, the differentially expressed proteins were subjected to analysis with the DAVID (the Database for Annotation, Visualization, and Integrated Discovery; version 2021) webware for annotation enrichment analysis [66].

The quantitative phosphoproteomic information was extracted from the Proteome Discoverer results and uploaded to phosphomatics webware [67] to perform the kinase-substrate enrichment analysis and annotation enrichment analysis. KSEA analysis was performed with the PhosphoSitePlus and NetworKIN datasets, using a NetworKIN score cutoff of 2, a *p*-value cutoff of 0.05, and a substrate count cutoff of 5 [68]. Graphs were plotted using GraphPad Prism (version 9, GraphPad, San Diego, CA, USA) unless otherwise specified.

## Figures and Tables

**Figure 1 ijms-25-01047-f001:**
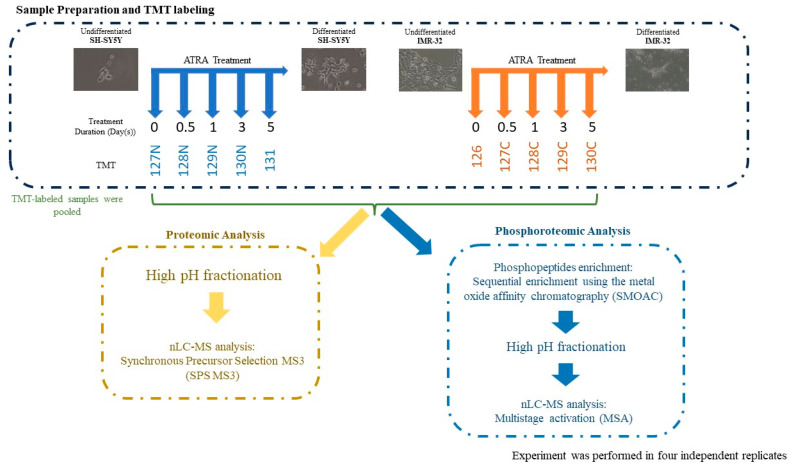
Experimental workflow. SH-SY5Y and IMR-32 cell lines were induced to differentiate by treating with ATRA over a 5-day period. Cells were harvested at the indicated time points, and samples were labeled with TMT-10plex reagents. The labeled samples were pooled and underwent proteomic and phosphoproteomic analysis. All experiments were conducted with four independent biological replicates.

**Figure 2 ijms-25-01047-f002:**
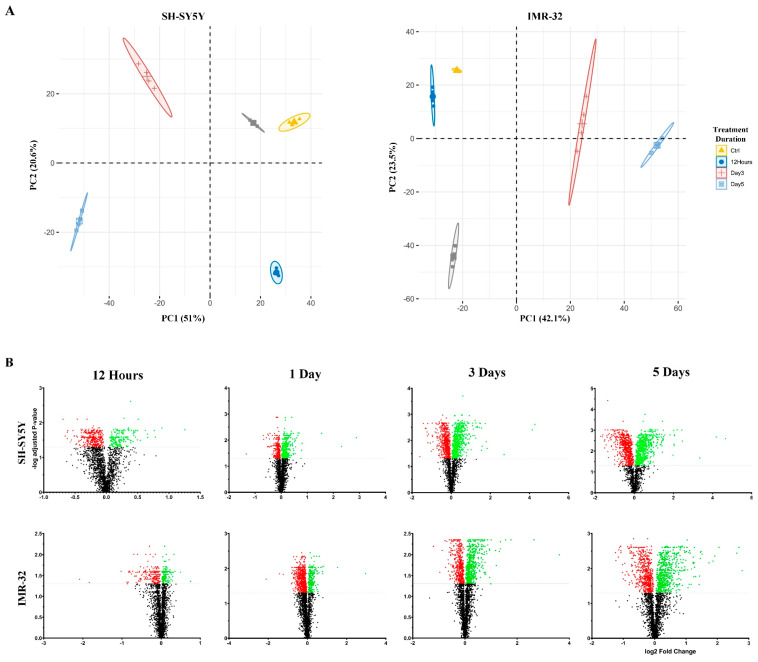
Proteomic profiles of SH-SY5Y and IMR-32 cells during all-*trans*-retinoic acid (ATRA)-mediated neuronal differentiation. (**A**) Principal component analysis (PCA) of the 20 samples (5 time points, 4 replicates each) depicting variance explained by the first 2 principal components. (**B**) Volcano plots of the TMT-labeled proteomic analysis in SH-SY5Y (upper panel) and IMR-32 (lower panel) at the time points indicated time points. Proteins that are not differentially expressed are represented in black, while upregulated and downregulated proteins are shown in red and green.

**Figure 3 ijms-25-01047-f003:**
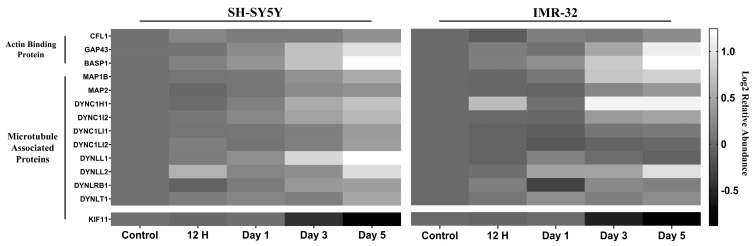
Heatmaps of relative protein expression level for cytoskeletal organization-related proteins. Expression values are represented relative to a control sample, set to a value of 1. Higher and lower expression levels are indicated by lighter and darker shades of grey, respectively.

**Figure 4 ijms-25-01047-f004:**
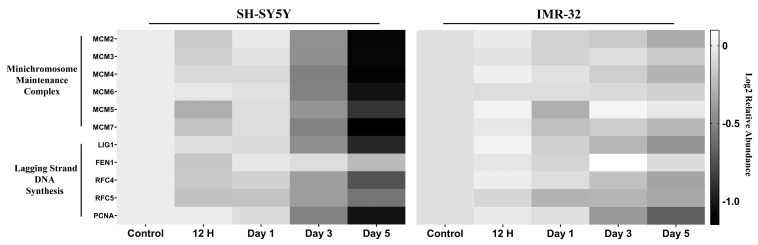
Heatmaps of relative protein expression level for DNA-replication-related proteins. Expression values are represented relative to a control sample, set to a value of 1. Higher and lower expression levels are indicated by lighter and darker shades of grey, respectively.

**Figure 5 ijms-25-01047-f005:**
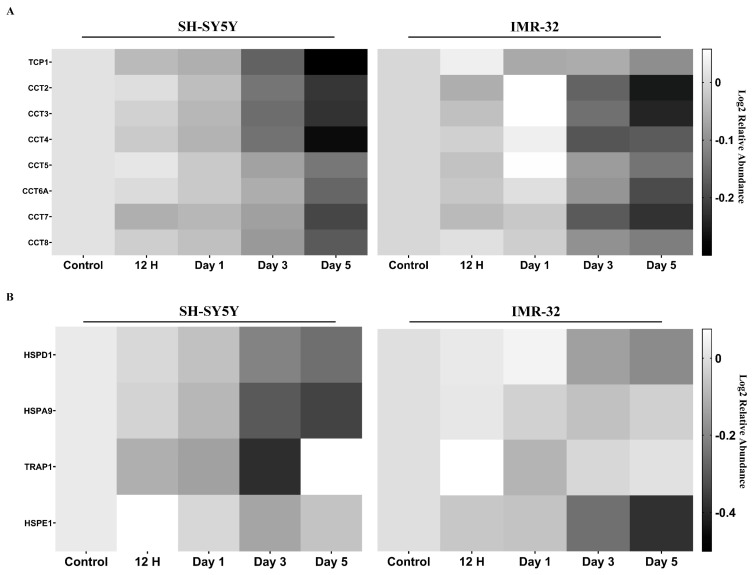
Heatmaps of relative protein expression level for (**A**) members of the T-complex protein ring complex and (**B**) mitochondrial chaperones. Expression values are represented relative to a control sample, set to a value of 1. Higher and lower expression levels are indicated by lighter and darker shades of grey, respectively.

**Figure 6 ijms-25-01047-f006:**
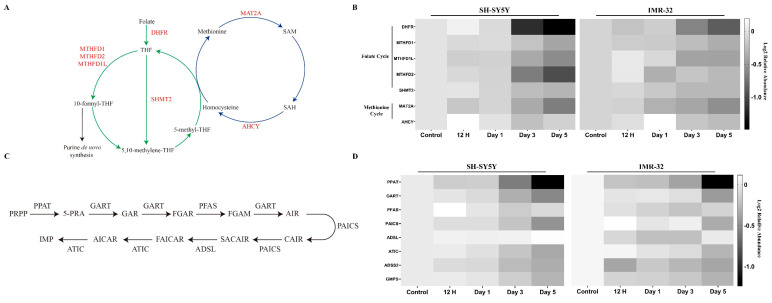
(**A**) Schematic of folate cycle (green) and methionine cycle (blue). Differentially expressed enzymes are highlighted in red. (**B**) Heatmaps of relative protein expression level for folate and methionine cycle-related proteins. (**C**) Schematic of de novo purine synthesis. (**D**) Heatmaps of relative protein expression level for de novo purine-synthesis-related proteins. Expression values of the heatmaps are represented relative to a control sample, set to a value of 1. Higher and lower expression levels are indicated by lighter and darker shades of grey, respectively.

**Figure 7 ijms-25-01047-f007:**
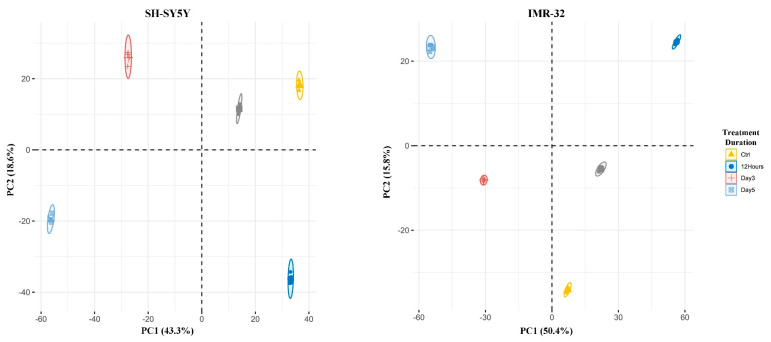
Phosphoproteomic profiles of SH-SY5Y and IMR-32 cells during ATRA-induced neuronal differentiation. Principal component analysis (PCA) of 20 samples (5 time points, 4 replicates each) depicting variance explained by the first 2 principal components.

**Figure 8 ijms-25-01047-f008:**
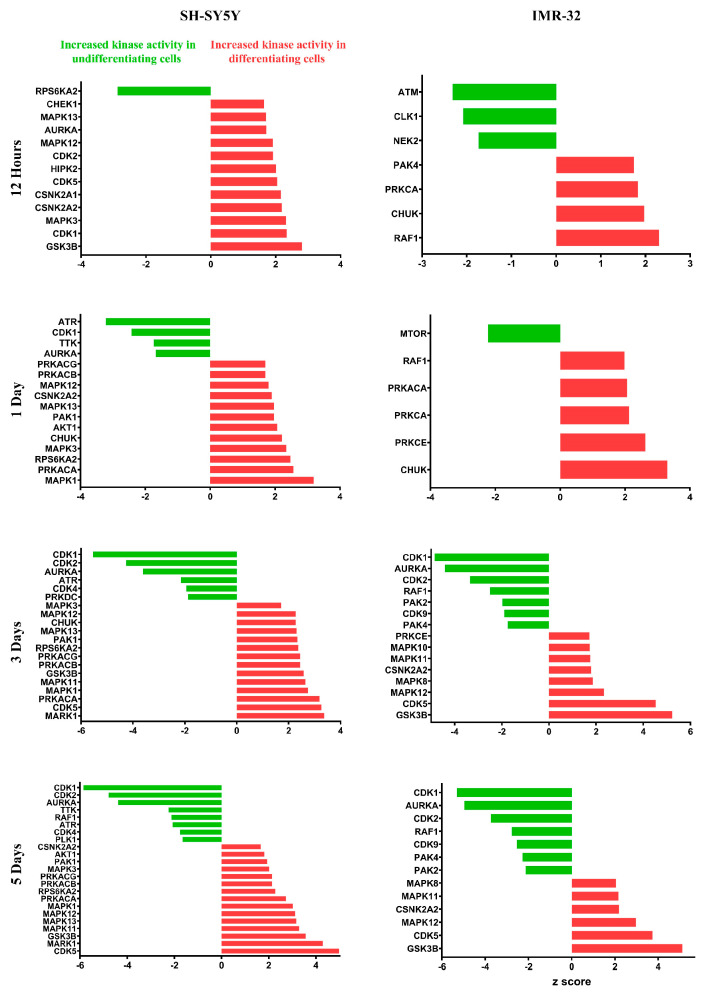
Kinase-substrate enrichment analysis (KSEA) of differentially expressed phosphopeptides. The analysis scored each kinase based on the relative hyper- or hypophosphorylation of its substrates. Positive and negative z-scores represent increased kinase activity in differentiating and control cells, respectively. Increased and decreased kinase activity is shown in red and green, respectively. Analysis was performed with the PhosphoSitePlus and NetworKIN datasets, using a NetworKIN score cutoff of 2, a *p*-value cutoff of 0.05, and a substrate count cutoff of 5.

## Data Availability

The mass spectrometry proteomics data have been deposited to the ProteomeXchange Consortium via the PRIDE [69] partner repository with the dataset identifiers PXD046900 and PXD046871.

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
