# Peer review of "Temporal Quantitative Proteomic and Phosphoproteomic Profiling of SH-SY5Y and IMR-32 Neuroblastoma Cells during All-Trans-Retinoic Acid-Induced Neuronal Differentiation"

_ijms, 2024, doi:10.3390/ijms25021047_

Round 1
Reviewer 1 Report
Comments and Suggestions for Authors
The study by Leung et al aims to provide data of temporal and quantitative profiling of the proteomes of two human neuroblastoma cell lines through treatment with ATRA as in the model of neuronal differentation. The study involved SH-SY5Y and IMR-32 cell lines The authors controlled several different time points The study is well designed and conducted; however, there are several issues that need to be addressed.
1.the materials and methods section needs to be rewrited with details regarding cell culture, treatment, reagents and so on. at this point, the reader is not able to repeat the experiment
2. the results need to be supported with Western Blot results
3. several concentrations of ATRA should be tested
4. nice graphical summary for the study would be nice
5. please mention the limitations of the study
Reviewer 2 Report
Comments and Suggestions for Authors
In this paper, you have presented interesting results concerning the induction of differentiation of SH-SY5Y and IMR-32 neuroblastoma cells by ATRA and the proteomic and phosphoproteomic analysis during the differentiation process. The information you have gathered is enormous, and it is difficult to connect all the changes in the levels of proteins and phosphoproteins to the differentiation process. Despite this difficulty, you have focused on a few important processes necessary for a neoplastic cell undergoing differentiation, mainly the cytoskeletal structure, cell cycle and metabolic status. The experiments performed were well designed, described in detail, well presented and analyzed, and thoroughly discussed. Knowledge of the levels of proteins and phosphoproteins is a requirement for further understanding the differentiation process of cancer cells, and I might suggest you would focus on the decision-making proteins (like commitment in leukemic cells) in the near future.
Minor corrections (suggestions):
in vivo, in vitro italics throughout the text
line 450: 37oCà37oC
line 451: CO2àCO2
line 452: all-trans àall-trans and throughout the text
line 453: dosesàconcentrations
line 455: peràaccording to
line 458: media was aspiratedàmedia were aspirated
line 463: 60oCà60oC
line 545: viaàvia
Reviewer 3 Report
Comments and Suggestions for Authors
In this manuscript, the cell lines SH-SY5Y and IMR-32 can be differentiated into neuron-like phenotypes through treatment with all-trans retinoic acid (ATRA)
Here, authors performed relative quantification of the proteomes and phosphoproteomes of SH-SY5Y and IMR-23 32 cells at different time points (before differentiation induction as well as during ATRA-induced differentiation).
Comments:
There are errors on the symbols and units of measurement, also some full abbreviations are missing in the text.
Information on many reagents and conditions of use is missing, for example, how all-trans retinoic acid is solubilized is not described.
Two cell lines are used in the manuscript, but it is not explained why the authors made this choice. It should be described how the human neuroblastoma cell lines SH-SY5Y and IMR-32 differ and the characteristics and limitations of the two lines used.
In some sections (e.g., 4.3 Phosphopeptide Enrichment) the values of the samples used are missing.
Some figures are missing from the statistics.
The authors state that the enrichment analysis revealed that the upregulated proteins were predominantly associated with neuronal and nervous system development and cytoskeleton organization. The authors could include some photos of the cells to confirm that differentiation alters cytoskeletal protein expression.
The authors should discuss their data considering what has already been reported in the literature on neuroblastoma cell differentiation.
The authors need to explain and justify why they conducted this study on neuroblastoma. The involvement of methionine and folate metabolism should be discussed more.
The folate cycle is necessary for methionine synthesis. Can this dysregulation affect the proliferation/reprogramming of neuroblastoma cancer cells?
Do the authors believe that there may be a basis for therapeutic intervention?
The work seems well conducted, but it is unclear why the authors conducted these tests and what the purpose of the work is.
The manuscript consists of a descriptive data set with very little discussion in the context of the challenges and the applications of neuroblastoma study.
It is not explained what the scientific application of the data and observations obtained might be.
Comments on the Quality of English Languagefine
Reviewer 4 Report
Comments and Suggestions for Authors
The manuscript by Leung et al., is a comprehensive proteomic and phosphoproteomic analysis of protein mediators of neuronal differentiation in cell culture. The author’s conclude “The current study compiled a comprehensive resource detailing the temporal abundance changes of thousands of proteins and phosphopeptides. These quantitative proteomic and phosphoproteomic profiles of differentiating SH-SY5Y and IMR-32 cells may serve as a useful reference for future studies utilizing these widely employed in vitro models of human neurons.” In my opinion, the data set fully support this conclusion. Overall, the manuscript is well written and the discussion of the highlighted changes is robust and appropriate. The use of TMT proteomics and phospho-peptide enrichment is well designed and presented. Overall, there are only a few minor comments to be addressed.
1. The normalization in the heat maps and their presentation, while correct, are a little counterintuitive to me. They are setting the highest abundance day at 100% (Green) and then color shift to red for those days where there is a decrease. I wonder if it would be more clear to set the control day at 1 and then use a heat map to show fold-change increases and decreases from 1. Additionally, while this could be done using a red green scale (e.g. 1 = yellow, fold decrease to green and increase to red) I think a gray scale would be more inclusive to individuals who are color blind (or a color-blind neutral palate). Regardless of the approach, showing an increase/decrease from control would be better representative of their question of how differentiation impacts gene expression from a non-differentiated state. At the very least, mentioning that the highest abundance was set to 1, green, should be more clearly stated.
2. The type of replicate is unclear. In the methods or on Figure 1, there should be a mention of if these are biological replicates or technical replicates (e.g. a single sample run 4 separate times through the MS). Additional details in what a replicate is should be mentioned.
3. In Table S5 for the GO terms, it is unclear what the different text blocks are. How are these separated by up/down regulation and cell type (Cell type may be inferred by population total, so the left set of blocks is SH-SY5Y and right is IMR?) Also, one of the blocks on the left side does not have a Fold-Enrichment column.
4. The discussion on CDKs is interesting and consistent with literature. The author’s may also mention that CDK5, unlike the other CDKs, is not regulated by cyclins, rather by p35/p25. Further supporting a unique effect on this non-cyclin CDK.
5. Some of the figures are very difficult to read. This is most likely due to making the PDF and the journal does not include original images. For instance, I cannot read any of the text in figure 2, even when increasing the magnification it is blurred.
6. There are a couple of lines that were part of the template document. For instance, lines 97-99 state “This section may be divided by subheadings. It should provide a concise and precise description of the experimental results, their interpretation, as well as the experimental conclusions that can be drawn.” Also section 6 patents should be deleted.
7. In the statistics section, the statement “Graphs were plotted using GraphPad Prism (version 9) unless otherwise specified.” Is mentioned twice.
Round 2
Reviewer 1 Report
Comments and Suggestions for Authors
The mamuscript has been revised succesfully.
Reviewer 3 Report
Comments and Suggestions for Authors
The authors addressed most of the required comments and changed the manuscript accordingly. The manuscript sounds now better.
Comments on the Quality of English Languageenglish is good